# IMAC: IMPLICIT MOTION-AUDIO COUPLING FOR CO-SPEECH GESTURE VIDEO GENERATION

## ABSTRACT

Co-speech gestures are essential to non-verbal communication, enhancing both the naturalness and effectiveness of human interaction. Although recent methods have made progress in generating co-speech gesture videos, many rely on explicit visual controls, such as pose images or TPS keypoint movements, which often lead to artifacts like inconsistent backgrounds, blurry hands, and distorted fingers. In response to these challenges, we present the Implicit Motion-Audio Coupling (IMAC) method for co-speech gesture video generation. IMAC strengthens audio control by coupling implicit motion parameters, including pose and expression, with audio inputs. Our method utilizes a two-branch framework that combines an audio-to-motion generation branch with a video diffusion branch, enabling realistic gesture generation without requiring additional inputs during inference. To improve training efficiency, we propose a two-stage slow-fast training strategy that balances memory constraints while facilitating the learning of meaningful gestures from long frame sequences. Furthermore, we introduce a large-scale dataset designed for co-speech gesture video generation and demonstrate that our method achieves state-of-the-art performance on this benchmark.

## 1 INTRODUCTION

Co-speech gestures are essential to non-verbal communication, playing a key role in how humans convey meaning. As virtual agents become more interactive, generating gestures synchronized with speech is crucial for human-computer interaction. While methods exist for co-speech gesture generation (Liu et al., 2024; Chen et al., 2024) and talking head videos (Tian et al., 2024; Xu et al., 2024a), generating realistic co-speech gesture videos remains challenging, especially for half-body animations that mimic natural human communication. Addressing this is critical for improving virtual agents in content creation, entertainment, and education.

Several methods have been proposed to address the co-speech gesture video generation task. For instance, MYA (Huang et al., 2024) and Vlogger (Corona et al., 2024) utilize motion generation models to create motion data, which is then rendered as pose image sequences to serve as explicit visual control. These pose image sequences, along with a reference image, are then used as input to a diffusion model to generate the final videos. Similarly, S2G (He et al., 2024) uses TPS keypoint movements as explicit visual control for gesture video generation. However, these approaches often struggle with issues such as inconsistent backgrounds, blurry hands, and distorted hands. The root cause of these artifacts lies in the explicit visual control, which edits the reference image. When this manipulation is translated into pixel space, the resulting distortions introduce visual artifacts that degrade the overall quality of the generated videos.

A straightforward solution to the above issues might be directly using audio information as the condition for video diffusion generation, similar to prior works in talking head generation (Tian et al., 2024; Xu et al., 2024a). However, unlike talking head generation, co-speech gesture video generation involves producing upper-body gestures. The many-to-many mapping between audio and gestures complicates this approach, as the correlation between audio and gestures is relatively weak. This makes it difficult for the weak audio signal alone to serve as an effective condition for generating co-speech gesture videos. To this end, we propose our Implicit Motion-Audio Coupling (IMAC) method to address the co-speech gesture video generation task. Inspired by Handiffuser (Narasimhaswamy et al., 2024), we couple implicit motion parameters (pose and expression)

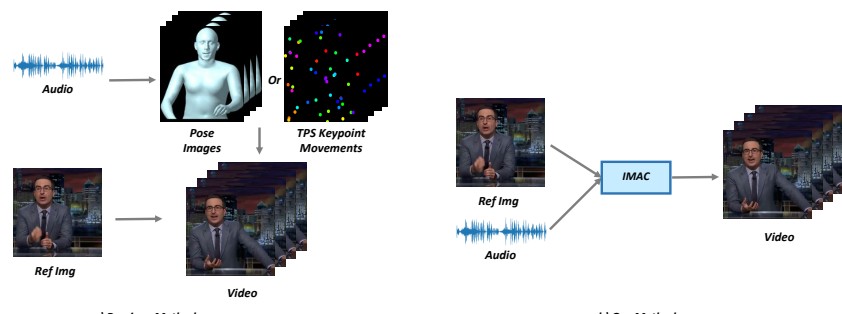

Figure 1: Comparison with previous methods. Previous methods typically use audio to generate explicit visual controls like pose images or TPS keypoint movements to drive the reference image and create a gesture video. In contrast, our method leverages implicit motion-audio coupling to directly drive the reference image.

with audio information to strengthen the model's audio control capability. Specifically, we employ a two-branch framework. In addition to the video diffusion branch, we introduce an audio-to-motion generation branch, which integrates audio information with motion parameters during training. This design allows the model to generate motion parameters that are closely aligned with the audio input while eliminating the need for additional inputs during inference.

Training such a model is challenging, as generating meaningful gestures requires long epochs and processing extended frame sequences. To address this, we propose a two-stage slow-fast training strategy. In the first stage, we train the audio-to-motion branch with long epochs and frame sequences. In the second stage, we jointly train the video diffusion and audio-to-motion branches, using short frame sequences for the video diffusion branch while continuing to feed long sequences into the audio-to-motion branch. This approach balances effective gesture learning with memory constraints.

We also introduce a new large-scale dataset for co-speech gesture video generation, with detailed information provided in the Experiment section. We summarize our contributions as follows: 1) We propose the Implicit Motion-Audio Coupling (IMAC) method, which couples implicit motion parameters with audio to enhance weak audio control and avoid artifacts caused by explicit visual control. 2) We introduce a two-branch framework that integrates an audio-to-motion generation branch alongside the video diffusion branch, enabling realistic gesture video generation without additional inputs during inference. 3) We present a two-stage slow-fast training strategy that balances memory constraints while learning meaningful gestures from long frame sequences. 4) We contribute a large-scale dataset for co-speech gesture video generation and verify the effectiveness of our method on it.

## 2 RELATED WORK

**Co-speech Gesture Video Generation.** A common approach to co-speech gesture video generation splits the process into two stages: audio-to-motion generation and motion-to-video synthesis. Speech2Gesture (Ginosar et al., 2019) uses a generative adversarial networks (GAN) (Goodfellow et al., 2020) to generate 2D skeleton movements, followed by another GAN for video synthesis. Speech-Drives-Templates (Qian et al., 2021) applies a variational autoencoders (VAE) (Kingma, 2013) in the first stage and image warping in the second. Vlogger (Corona et al., 2024) utilizes two diffusion models to generate pose images as explicit visual controls and the corresponding human videos. MYA (Huang et al., 2024) is originally designed as a pose image guided video generation framework but can be easily adapted for co-speech gesture video generation by integrating an audio-to-motion technique. S2G (He et al., 2024) employs a diffusion model to map audio to keypoint movements, using a nonlinear thin-plate spline (TPS) transformation to decouple latent motion from video. However, these methods often suffer from artifacts such as inconsistent backgrounds, and blurry, or distorted hands. These issues arise from the explicit visual control, which modifies the reference image and introduces distortions when translated into pixel space. In contrast, our method couples implicit motion parameters with audio information, enabling the generation of high-quality gesture videos without such artifacts.

**Co-speech Gesture Generation.** Co-speech gesture generation aims to produce lifelike human gestures synchronized with given audio inputs. Due to the complex, many-to-many relationship between audio and gestures, generative models have proven more effective than deterministic models. Researchers have applied various generative approaches to this task. For example, GAN-based models have been used to predict skeleton movements, enhancing gesture diversity (Ginosar et al., 2019). Other methods have employed VAE (Yi et al., 2023; Liu et al., 2024) and flow-based models (Ye et al., 2022) to capture the intricate relationship between audio and gestures. Recently, diffusion-based generative models have gained attention, with several studies exploring their use in gesture generation (Zhu et al., 2023; Yang et al., 2023a; Chen et al., 2024). However, these approaches typically generate only motions, which are later rendered as pose images rather than gesture videos. In contrast, our work introduces an audio-to-motion generation branch that directly generates implicit motion parameters without rendering, using them to enhance the model's audio control capabilities for co-speech gesture video generation.

**Talking Head Video Generation.** A related area of research is talking head video generation, where key challenges include achieving precise synchronization between lip movements and audio while maintaining the subject's visual realism. Researchers have tackled these challenges using various innovative techniques. Recent advancements have utilized large-scale pre-trained diffusion models, integrating them with specialized audio control modules to achieve impressive results in both visual fidelity and audio-visual synchronization (Tian et al., 2024; Stypułkowski et al., 2024; Xu et al., 2024a). Inspired by these works, we also employ audio as a direct control signal for co-speech gesture video generation. However, since audio alone is often too weak to effectively drive gesture video generation, we couple it with implicit motion parameters to enhance the model's control capabilities.

## 3 METHOD

### 3.1 PRELIMINARY: LATENT DIFFUSION MODELS (LDMS)

Latent Diffusion Models (LDMs) (Rombach et al., 2022; Blattmann et al., 2023b) are effective and efficient approaches for generating high-quality images or videos by performing diffusion and denoising in the latent space. Compared to pixel-level diffusion models (Ramesh et al., 2022; Song et al., 2021; Ho et al., 2022), LDMs significantly reduce computational complexity by operating in a compressed latent space derived from a pre-trained VAE.

Given an input video $x \in \mathbb{R}^{F \times H \times W \times 3}$, a VAE encoder $\mathcal{E}$ compresses it into a latent representation $z = \mathcal{E}(x)$, where $z \in \mathbb{R}^{F \times h \times w \times c}$, with $h < H$ and $w < W$ denoting the spatially downsampled dimensions and $c$ being the number of latent channels. To recover the original video, a VAE decoder $\mathcal{D}$ reconstructs the latent representations back into the pixel space $\bar{x} = \mathcal{D}(z)$.

The LDM framework consists of two primary processes: *diffusion* and *denoising*. During the *diffusion* phase, noise is progressively added to the latent variable $z$, resulting in a noisy latent representation $z_t$, where $t \in \{1, \ldots, T\}$ with $T$ being the total number of diffusion steps. The *denoising* phase applies a learned denoising function $\epsilon_\theta(z_t, t, c)$ to iteratively remove the noise and recover the clean latent representation $z_0$. The denoising model is typically optimized using the following objective:

$$\mathcal{L}_{\text{LDM}} = \mathbb{E}_{z, \epsilon \sim \mathcal{N}(0,1), t} \left[ \|\epsilon - \epsilon_\theta(z_t, t, c)\|_2^2 \right], \tag{1}$$

where $\epsilon$ represents the added noise, and $c$ represents any conditional guidance, such as text prompts or initial frames. The model typically adopts a 3D U-Net architecture (Wang et al., 2023b;a) for improved temporal modeling, particularly when dealing with video data.

In this work, we build our model on the open-sourced Image-to-Video (I2V) diffusion model, I2VGen-XL (Zhang et al., 2023b), which is capable of generating complex motions from a single input image (Xing et al., 2023; Lin et al., 2024). While our model primarily utilizes I2VGen-XL for dynamic video generation, it remains flexible and can also integrate other video diffusion models, such as SVD (Blattmann et al., 2023a).

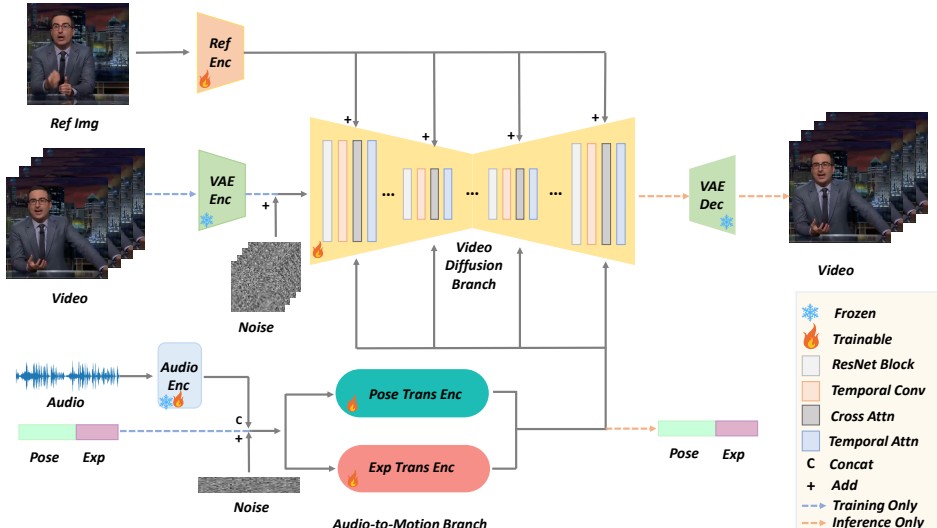

Figure 2: Overview of our method. Video frames are encoded using a VAE encoder and fused with multi-frame noise, after which the video diffusion model performs the denoising process for video generation. A reference image is encoded using a reference encoder, and the extracted features are added at the position following the cross-attention module in the video diffusion model. Motion parameters, including pose and expression, are concatenated with the encoded audio features. After being fused with noise, pose and expression encoders are employed for the denoising process to generate motion parameters. The resulting motion parameter representation is then passed to the video diffusion model through cross-attention. Note that a frozen pretrained Hubert model and a trainable temporal self-attention layer are utilized during the audio encoding process. A more detailed explanation is provided in the method section.

## 3.2 PROBLEM DEFINITION

Given a human reference image $I_{\text{ref}}$ and an audio sequence $A \in \mathbb{R}^F$, where $F$ denotes the number of frames, co-speech gesture video generation seeks to produce a video that faithfully retains the appearance of the individual in $I_{\text{ref}}$ while synchronizing fluidly with the audio sequence $A \in \mathbb{R}^F$. The goal is to generate a video that not only preserves the visual identity of the reference image but also captures the nuances of the speaker's style in the audio. This results in natural, expressive gestures closely tied to the audio cues, creating a cohesive and lifelike video.

## 3.3 NETWORK PIPELINES

### 3.3.1 REFERENCE ENCODER

In human-centered video generation, reference control is essential for preserving fine-grained appearance details across frames, such as facial identity, clothing textures, and background elements. This ensures consistency between the reference image and the generated video. Previous methods like ControlNet (Zhang et al., 2023a) fall short because they rely on control features (e.g., depth and edge maps) that are spatially aligned with the target image. Such reliance does not accommodate the spatial misalignment between reference and target images inherent in our task. Similarly, methods like ReferenceNet (Hu, 2024; Chang et al., 2023) introduce computational inefficiencies by employing complex attention mechanisms to handle reference features, which significantly increase the computational load.

In co-speech gesture generation, we focus on a few identities since learning gestures for each identity requires substantial data, and scaling to more identities significantly increases data needs. Therefore, using a heavy encoder is unnecessary for this setting. To address this, we utilize a lightweight reference encoder composed of a series of residual-connected convolutional modules to extract reference features. More importantly, the reference encoder adjusts the feature dimensions to match those of the noise latent, allowing the features to be added together for further learning. In addi-

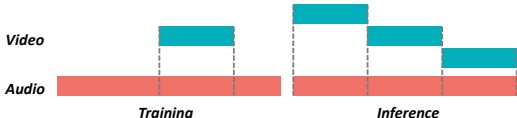

Figure 3: Illustration of our slow-fast training and inference process. During the second training stage, audio is processed over a longer frame sequence, while the video is trained on a randomly selected shorter continuous frame sequence. Note that only the corresponding audio frames are passed to the video generation model. During inference, the model generates the first short video clip based on the given reference image, and for subsequent video clips, the last frame of the previous clip is used as the new reference image.

tion, I2VGen-XL utilizes the VAE and CLIP encoders to extract low-level and high-level features from the reference image, respectively. These features are integrated into the video diffusion model through cross-attention, and we preserve this component in our approach. For clarity and simplicity, we omit this part in Fig. 2.

### 3.3.2 AUDIO-TO-MOTION GENERATION

Current diffusion-based methods often rely on explicit visual control for generating co-speech gesture videos, typically using an audio-to-motion framework that first generates pose image sequences (Huang et al., 2024) or TPS keypoint movements (He et al., 2024). This introduces system complexity and results in suboptimal output quality. A more straightforward approach would be to directly condition the diffusion model on audio information. However, relying solely on audio is insufficient due to the weak correlation between audio and gestures. To address this, we propose enhancing the audio input with implicit motion information. Specifically, we incorporate SMPL-X (Pavlakos et al., 2019) motion parameters, comprising poses and expressions, into our training process, and subsequently generate the corresponding motion parameters during inference. Formally, given an audio sequence $A \in \mathbb{R}^F$, the model generates a motion parameter sequence, including a pose vector $\beta \in \mathbb{R}^{F \times 3J}$ for joint rotations, capturing finger articulation across $J$ joints, and an expression vector $\theta \in \mathbb{R}^{F \times 100}$ for facial expressions.

**Audio Encoding**

Given an audio sequence $A \in \mathbb{R}^F$, we extract the MFCC feature and process it with one temporal self-attention layer. Concurrently, we use a pre-trained HuBERT (Hsu et al., 2021) model to extract the semantic feature. Afterward, two types of features are concatenated and fused by a linear layer.

**Motion Parameter Generation**

Given the extracted audio features, a conventional approach is to use a cross-attention layer to condition motion parameter generation on these audio features. However, our primary objective is video generation, and relying on cross-attention diminishes the strength of the audio information in the subsequent video generation process. To address this limitation, we propose concatenating the audio features with noisy motion parameters during training to retain the strength of the audio signal.

For processing pose and expression information, we employ two transformer encoders, each comprising a series of temporal self-attention layers to handle the respective features. Once the motion features are encoded, the resulting representation is passed to the video generation branch via cross-attention, ensuring effective integration of the motion data into the video synthesis.

It is important to note that we do not use classifier-free guidance (Ho & Salimans, 2022) in our framework, as our goal is to enhance the audio information using motion parameters, not to introduce variability in motion generation. Moreover, the video diffusion branch already incorporates diversity, and adding diversity to the audio-to-motion branch would significantly complicate training.

### 3.4 SLOW-FAST TRAINING AND INFERENCE

#### 3.4.1 TRAINING STRATEGY

Training our network presents significant challenges. First, audio-to-motion generation generally requires extended training epochs (e.g., 3000), but training video diffusion models over such pro-

longed epochs can be time-consuming. Second, video diffusion models typically utilize a limited number of frames (e.g., $F = 16$) due to memory constraints; however, this is insufficient for capturing meaningful gestures, which often last longer. To address these challenges, we propose a two-stage slow-fast training strategy.

In the first stage, we focus exclusively on training the audio-to-motion branch with an extended sequence of frames ($F = 80$). The loss function at this stage is similar to Eq. 1, with motion denoted as $m$:

$$\mathcal{L}_{\mathrm{M}} = \mathbb{E}_{\boldsymbol{m},\epsilon \sim \mathcal{N}(0,1),t} \left[ \|\epsilon - \epsilon_\theta(\boldsymbol{m}_t, t, \boldsymbol{c})\|_2^2 \right], \qquad (2)$$

where $\mathcal{L}_{\mathrm{M}}$ represents the audio-to-motion branch loss.

In the second stage, we inherit the weights from the audio-to-motion branch and initialize the video diffusion model using the pre-trained I2VGen-XL model (Zhang et al., 2023b). During this stage, we continue using the longer frame sequences ($F = 80$) for the audio-to-motion branch while randomly selecting a continuous sequence of shorter frames ($F = 16$) for the video diffusion branch. The training losses at this stage are:

$$\mathcal{L} = \mathcal{L}_{\mathrm{LDM}} + \mathcal{L}_{\mathrm{M}}, \qquad (3)$$

where $\mathcal{L}_{\mathrm{LDM}}$ denotes the loss for the video diffusion model.

This two-stage strategy allows us to effectively train the model while managing memory constraints.

### 3.4.2 INFERENCE

During inference, we employ a strategy similar to that used in training. Given an audio sequence, we divide it into longer frame sequences ($F = 80$) and sequentially pass shorter chunks ($F = 16$) to the video diffusion model. For long video generation, previous methods often use complex schemes, such as overlap denoising (Huang et al., 2024) or optimal motion selection (He et al., 2024). However, in our experiments, we find that simply using the last frame of the previously generated clip as a reference image is sufficient.

An illustration of our slow-fast training and inference process is shown in Fig. 3.

## 4 EXPERIMENT

### 4.1 DATASET

Our objective is to construct a large-scale dataset for co-speech gesture video generation. To achieve this, we collect numerous gesture videos from YouTube and annotate them with corresponding labels. To minimize the costs and time associated with manual filtering and annotation, we developed an efficient pipeline that automates video filtering and produces high-quality annotations. Specifically, we focus on four individuals: Oliver, Noah, Seth, and Huckabee. Using identity labels, we automatically download videos from YouTube[1] and apply several processing steps, including filtering and annotation, as described below.

### 4.1.1 DATA PROCESSING

**Video Filtering.** Some of the collected video candidates may not meet the high-quality standards required for co-speech gesture video generation. For instance, certain videos may feature multiple individuals or exhibit significant scene changes across frames. To handle this, we first segment the videos into clips using SceneDetect[2], ensuring that videos with different scenes are separated. Next, we filter out multi-person videos by employing TalkNet (Tao et al., 2021), a speaker diarization model that detects and distinguishes different speakers in a video. By using TalkNet, we ensure that the remaining videos contain only single-person scenes. Finally, we use MediaPipe (Lugaresi et al., 2019) to detect human faces and discard videos with low face detection confidence, which are typically videos featuring side views of faces. Additionally, we retain only video clips longer than 3 seconds, as shorter clips are unlikely to contain meaningful gestures.

**Data Annotation.** After obtaining the video clips from the previous stage, we annotate our data as follows. First, we extract the audio from the videos using ffmpeg. Next, SHOW (Yi et al., 2023)

---

[1]https://github.com/yt-dlp/yt-dlp

[2]https://github.com/Breakthrough/PySceneDetect

Table 1: Data Statistics.

| Identity | Posture | Resolution | Duration / h | Train / h | Test / h |
|----------|---------|------------|--------------|-----------|----------|
| Oliver | Sitting | 720x1280 | 13.73 | 13.20 | 0.53 |
| Noah | Sitting | 1080x1920 | 10.73 | 10.20 | 0.53 |
| Seth | Sitting | 1080x1920 | 5.66 | 5.24 | 0.42 |
| Huckabee | Standing | 1080x1920 | 2.88 | 2.76 | 0.12 |
| All | - | - | 33.00 | 31.40 | 1.60 |

method is applied to reconstruct the holistic whole-body mesh, i.e., the SMPL-X motion parameters (including pose and expression). Since the videos are typically rectangular, we need to crop them into square frames. The key challenge during cropping is determining the optimal cropping position. To address this, we render the mesh parameters into mesh frames. Then, we binarize these mesh images to obtain segmentation masks. Using these masks, we compute the largest bounding box of the person in the video, crop the width based on this value, and pad the height accordingly to achieve a square aspect ratio. By ensuring that all frames in the same video use the same bounding box, we maintain a consistent camera position, which is crucial for our task, as modeling camera movement is challenging.

After processing the data, we obtain our final dataset and split it into training and test sets. The detailed statistics are provided in Table. 1.

## 4.2 METRICS

For better comparison with existing works, we follow the previous work (He et al., 2024) to design our evaluation metrics. We assess the quality, diversity, and synchronization between gestures and speech using four key metrics: Fréchet Gesture Distance (FGD) (Qian et al., 2021), Diversity (Div.) (Liu et al., 2022), Beat Alignment Score (BAS) (Li et al., 2021), and Fréchet Video Distance (FVD) (Unterthiner et al., 2018).

FGD quantifies the distributional discrepancy between real and synthesized gestures within the feature space. Diversity measures the average feature distance among generated gestures, indicating their variability. To compute both FGD and Diversity, we first extract 2D human skeletons using the DWPose (Yang et al., 2023b) framework, a readily available pose estimation tool, and train an auto-encoder using skeleton data from our training dataset. BAS represents the mean distance between the nearest speech beats and corresponding gesture beats, ensuring temporal coherence between speech and gestures. For this metric, skeletons are also extracted from the test set to identify the gesture beats. FVD assesses the overall fidelity of the gesture videos by leveraging the I3D (Wang et al., 2019) classifier, which is pre-trained on the Kinetics-400 (Kay et al., 2017) dataset, to compute FVD within the feature space.

We also conduct a user study to validate the qualitative performance of our model, which is introduced in the Appendix.

## 4.3 QUALITATIVE RESULTS

We present our results for four identities in Fig. 4. As shown, our model generates high-quality co-speech gesture videos that are both clear and consistent. Additional video results can be found in the supplementary material.

## 4.4 COMPARISONS

We compare our method with two open-source prior works, S2G (He et al., 2024) and MYA (Huang et al., 2024). For a fair comparison, we finetune their pre-trained models using our proposed dataset. Note that S2G is designed for co-speech gesture video generation, while MYA is focused on pose images guided video generation. Therefore, we first train an audio-to-motion model, DiffSHEG (Chen et al., 2024), on our dataset to generate the pose images for MYA. The results are presented in Fig. 5 and Table. 2. Video comparison results are included in the supplementary material. The results of the user study are introduced in the Appendix.

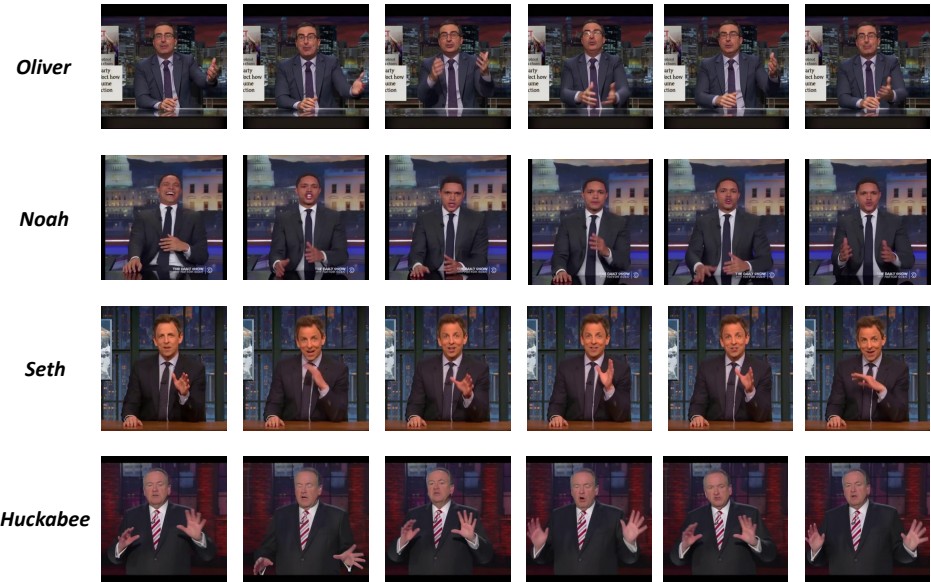

Figure 4: Qualitative results. From top to bottom, the identities are Oliver, Noah, Seth, and Huckabee. Given a reference image (the leftmost image) and an accompanying audio sequence (not depicted), our model generates high-quality co-speech gesture videos corresponding to the input. More video results are provided in the supplementary material.

Table 2: Quantitative comparison with previous works on four objective metrics. Bold text indicates the best performance.

| Model | FGD ↓ | Div. ↑ | BAS ↑ | FVD ↓ |
|---|---|---|---|---|
| S2G (He et al., 2024) | 3.69 | 180.59 | 0.7280 | 816.03 |
| MYA (Huang et al., 2024) | 24.24 | 224.14 | **0.7452** | 1823.97 |
| **Ours** | **2.01** | **240.67** | 0.7437 | **652.16** |

As illustrated in Fig. 5, our method generates high-quality videos without blurry hands or finger distortion and maintains a consistent background. In contrast, S2G and MYA struggle with inconsistent backgrounds and suffer from blurry hands and distorted fingers. It is important to note that the blur produced by our method is natural motion blur, whereas the blur in S2G and MYA is caused by their explicit visual TPS keypoint or pose image control. When this control is translated into pixel space, it results in blurred outputs. These flaws are even more pronounced in the videos, and we encourage readers to review the video comparisons available in the supplementary material. Additionally, MYA often memorizes appearance features during training. This causes the generated videos to replicate the memorized appearance instead of using the reference image, resulting in inconsistencies, as shown in Fig. 5.

The quantitative results are presented in Table. 2. As seen in Table. 2, our model achieves state-of-the-art performance across three objective metrics. In particular, the superior results on FGD and Diversity demonstrate our model's effectiveness in generating natural and diverse gestures. In addition, our model achieves the best FVD performance, indicating its ability to generate higher-quality videos compared to previous works. The lower BAS performance of our method compared to MYA may be due to the audio-to-motion generation stage we trained for MYA. While it does not produce ideal gestures, it effectively captures the alignment between gesture peaks and audio peaks, which positively impacts the BAS metric.

### 4.5 ABLATION ANALYSIS

**Without Reference Encoder.** To assess the effectiveness of our proposed reference encoder, we experiment by removing it during training and relying solely on the VAE and CLIP encoders for extracting reference image features. As shown in Fig. 6, the generated video exhibits background

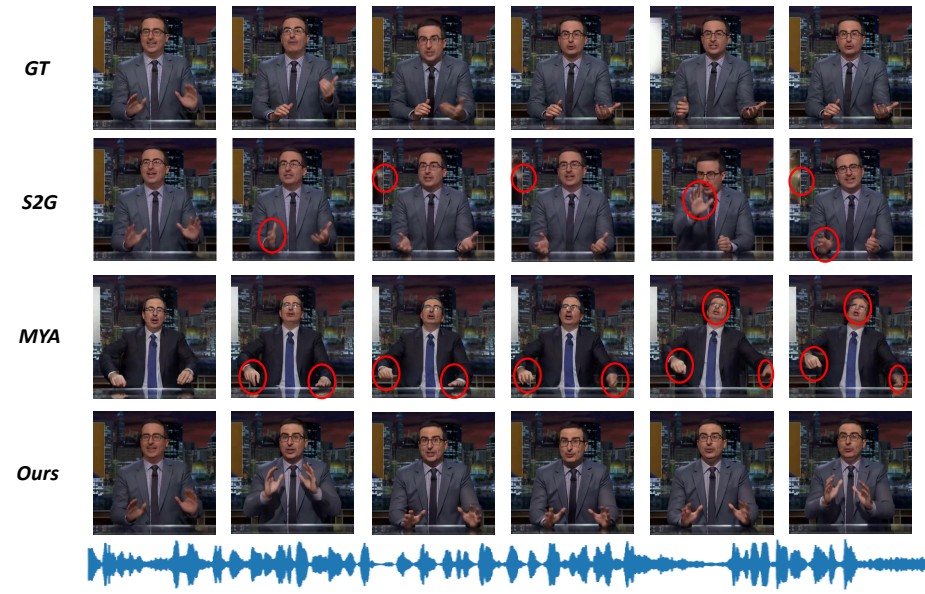

*Unless every single one of her emails was just a JPEG of a dog dressed as Dracula, in which case, yeah, you know what, that is kind of fun.*

Figure 5: Qualitative comparison with previous works. The leftmost image in the GT column is used as the reference image. Red circles highlight the obvious flaws in previous methods. As shown, prior works struggle with issues such as inconsistent backgrounds, blurry hands, and distorted fingers. In contrast, our method generates high-quality videos without these artifacts. Video results are available in the supplementary material.

Table 3: Quantitative ablation study on four objective metrics. Bold text indicates the best performance.

| Model | FGD ↓ | Div. ↑ | BAS ↑ | FVD ↓ |
|---|---|---|---|---|
| w/o Ref | 14.27 | 194.73 | 0.7419 | 1513.81 |
| w/o Motion | 21.95 | 189.02 | 0.7400 | 2406.91 |
| w/o First Stage | 48.86 | 180.60 | 0.7373 | 2373.28 |
| w/o Slow-Fast | 12.27 | 199.11 | 0.7411 | 1314.56 |
| **Ours** | **2.01** | **240.67** | **0.7437** | **652.16** |

inconsistencies with the reference image, and the color of the clothes is noticeably different. Additionally, the removal of the reference encoder degrades visual quality, resulting in a lower FVD score as shown in Table. 3. This decline in visual quality also compromises pose estimation accuracy, leading to lower performance on other objective metrics. These results emphasize the importance and effectiveness of the reference encoder in maintaining both visual consistency and overall quality.

**Without Motion Parameter Generation.** To evaluate the effectiveness of our proposed motion-audio coupling scheme, we experiment by removing the motion parameter generation and relying solely on audio information for guidance. For a fair comparison, we retain both pose and expression transformer encoders to process the audio features. As shown in Fig. 6, the generated video exhibits distorted hands, extra fingers, and hands that appear detached from the body. These artifacts indicate that relying only on audio results in a weak correlation between audio and video during training, which negatively impacts generalization during testing. Additionally, this configuration produces lower scores on objective metrics as shown in Table. 3, particularly in FVD, demonstrating that relying solely on weak audio information leads to poorer overall visual quality during testing.

**Without First Stage Training.** In our experiments, we follow a two-stage training strategy. Here, we evaluate the effect of removing the first stage and directly training the second stage. Skipping the first stage forces the model to focus more on the motion parameter generation branch during training, which in turn compromises the video generation capabilities. As shown in Fig. 6, the generated video exhibits issues such as extra fingers, and more critically, significant motion inconsistency. We

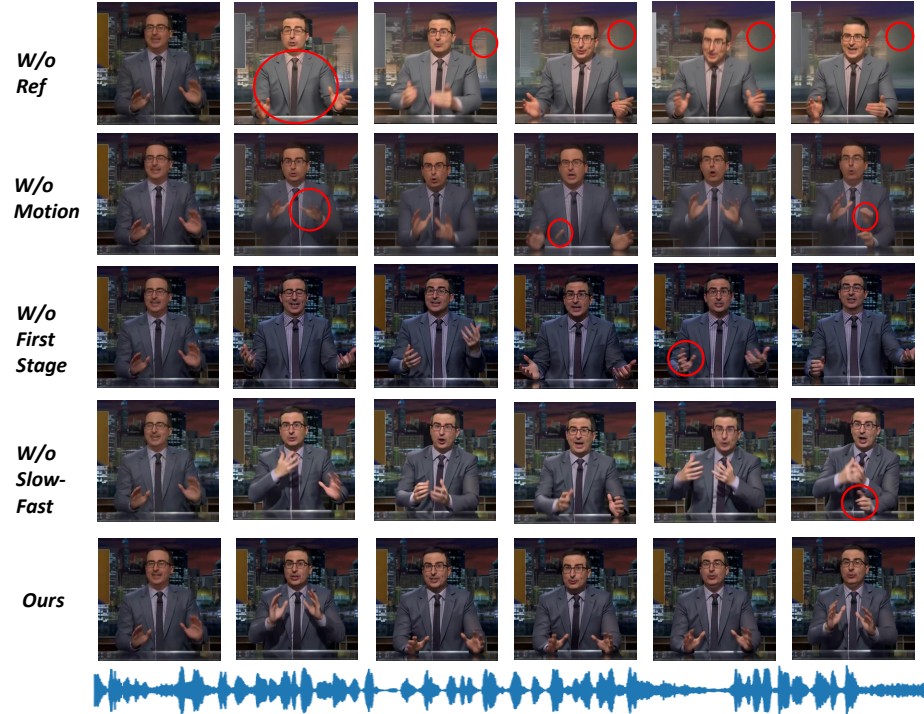

Figure 6: Qualitative ablation study. The leftmost image is used as the reference image. Red circles highlight the obvious flaws in incomplete settings. As shown, removing certain modules leads to issues such as inconsistent backgrounds, extra hands, and distorted or additional fingers. In contrast, our full method generates high-quality videos without these artifacts. Video results are available in the supplementary material.

encourage readers to view the supplementary videos to observe this flaw more clearly. Furthermore, training without the first stage yields the worst FGD and Diversity scores and results in the second-worst FVD, as shown in Table 3. This outcome suggests that bypassing the first stage leads to even poorer performance in coupling motion information than omitting motion entirely, indicating that the absence of first-stage training significantly hampers the video diffusion branch's performance.

**Without Slow-Fast Training.** To evaluate the effectiveness of our slow-fast training scheme, we train the audio-to-motion branch using shorter frame sequences ($F = 16$) instead of longer frame sequences ($F = 80$). As shown in Fig. 6, the generated results exhibit artifacts such as extra hands, and the motion shaking is severe. We encourage readers to watch the supplementary videos to better observe these issues. As indicated in Table 3, removing the slow-fast training scheme leads to lower performance across all objective metrics, further highlighting its importance in achieving smooth and realistic generation.

We also conduct a user study about ablation analysis, which is introduced in the Appendix.

## 5 CONCLUSION

In this paper, we introduce the Implicit Motion-Audio Coupling (IMAC) method, designed to tackle key challenges in co-speech gesture video generation by integrating implicit motion parameters with audio information. Our innovative two-branch framework, which combines an audio-to-motion generation branch with a video diffusion branch, facilitates realistic gesture generation without requiring additional inputs during inference. To optimize the training process, we propose a two-stage slow-fast training strategy, allowing the model to learn meaningful gestures efficiently while addressing memory constraints. Additionally, we develop a large-scale dataset tailored for co-speech gesture video generation and demonstrate the state-of-the-art performance of our method on this dataset. Extensive experiments and analysis confirm that our approach generates realistic, natural co-speech gesture videos that align seamlessly with the audio.

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

# A APPENDIX

## A.1 IMPLEMENTATION DETAILS

The training process is divided into two stages. In the first stage, we train only the audio-to-motion branch using audio and motion parameters. This stage runs for 3,000 epochs with a learning rate of 1e-4. The first stage is trained on 8 RTX 8000 GPUs for 1 day with a batch size of 256. In the second stage, we train the entire network for 200 epochs with a learning rate of 1e-5. This stage is trained on 4 A100 GPUs for 4 days with a batch size of 1. For the audio-to-motion branch, the frame sequence length is 80, while for the video diffusion branch, it is 16. The input image resolution is 512x512. The classifier-free guidance (CFG) scale is set to 3.5 for the video diffusion branch.

## A.2 USER STUDY

To further evaluate the visual performance of our method, we conduct a user study comparing the gesture videos generated by each method and each ablation study. We sample 30 generated videos from our test set for each method, and 20 participants are invited to rank the videos. Participants are asked to evaluate the videos based on four criteria:

**Identity Preservation**: Evaluates how well the essential characteristics and attributes of the human are maintained across the video.

**Visual Quality**: Assesses the video's clarity, with higher rankings indicating fewer issues such as blur, noise, and visual degradation.

**Temporal Consistency**: Measures frame-wise coherence, ensuring the logical progression of motion and visual elements across consecutive frames.

**Sound-Video Synchronization**: Judges the alignment between speech and gestures, assessing the accuracy of the generated motions.

Participants rank the videos, with rank 1 being the best. In comparison with previous works, the rankings are converted into points: rank 1 is assigned 3 points, rank 3 is given 1 point, and so on. For ablation studies, rank 1 is assigned 5 points, rank 5 is given 1 point, and so on. A higher overall score indicates better performance.

The user study results are presented in Table 4 and Table 5. As shown in Table 4, our method significantly outperforms others across all dimensions, demonstrating its ability to generate gesture videos with superior motion quality and overall visual fidelity. Although MYA achieves a slightly better BAS, it does not affect human perception of synchronization. Table 5 further highlights that our full model achieves state-of-the-art results in all metrics. The model without motion information performs the worst, which is consistent with the objective results shown in Table 3. The model without a reference encoder and the model without first-stage training yield comparable results, indicating that skipping the first-stage training shifts focus to the audio-to-motion branch while reducing the emphasis on the video diffusion branch, thereby degrading the visual quality. The model without slow-fast training achieves the second-best results but still falls short of our full model, demonstrating the effectiveness of our slow-fast training strategy.

We also show the user study interface in Fig. 7.

### A.2.1 STATISTICAL ANALYSIS

Given the limited number of participants, slight differences in rankings may not reliably indicate a significant preference for one method over another. To address this issue, we apply two statistical tests to validate the effectiveness of our user study, following Pakanen et al. (2022):

**Kruskal–Wallis Test.** We use the Kruskal-Wallis test to assess overall differences across multiple groups. This test is particularly robust when dealing with ordinal data and does not require a normal distribution, making it well-suited for our dataset. Since our user study data is ordinal and non-normally distributed, the Kruskal-Wallis test provides a reliable way to evaluate statistical significance. For more details on the calculation, readers can refer to the Wikipedia page[3]. The

---
[3]https://en.wikipedia.org/wiki/Kruskal–Wallis_test

Table 4: Quantitative comparison with previous works on four subjective metrics. Bold text indicates the best performance.

| Model | Preservation ↑ | Quality ↑ | Consistency ↑ | Synchronization ↑ |
|-------|---------------|-----------|---------------|-------------------|
| S2G | 2.15 | 2.12 | 2.18 | 2.18 |
| MYA | 1.10 | 1.22 | 1.13 | 1.13 |
| **Ours** | **2.75** | **2.66** | **2.69** | **2.70** |

test outputs a p-value, where a lower value indicates a higher degree of confidence in the observed differences across groups, signifying a stronger statistical significance.

**Dunn's Test.** Dunn's Test (Dunn, 1961) is a post-hoc test used for pairwise comparisons between groups. If the Kruskal-Wallis test indicates a statistically significant difference, Dunn's Test helps identify which specific groups differ from each other.

As shown in Table 6 and Table 7, the p-values are very low and approach zero, indicating substantial overall differences across the groups. To further analyze these differences, we refer to Fig. 8 and Fig. 9 for the results of Dunn's Test. In Fig. 8, all three groups show significant differences, validating the effectiveness of our user study. For instance, our method outperforms S2G by 0.5-0.6 points across all metrics (Table 4), and Dunn's Test confirms that the differences between the two groups are statistically significant, demonstrating that our method is superior and unaffected by the limited sample size. An interesting observation can be seen in Fig. 9, where the model without first-stage training and the model without the reference encoder show no significant difference, as indicated by a p-value of 1. This finding is consistent with the results in Table 5, where columns 1 and 2 yield similar scores. This suggests that it is difficult to determine which model performs better given the limited number of participants. However, this does not undermine the validity of the user study, as our full model demonstrates a statistically significant difference compared to all other incomplete models.

Table 5: Quantitative ablation study on four subjective metrics. Bold text indicates the best performance.

| Model | Preservation ↑ | Quality ↑ | Consistency ↑ | Synchronization ↑ |
|-------|---------------|-----------|---------------|-------------------|
| w/o Ref | 2.52 | 2.49 | 2.48 | 2.54 |
| w/o Motion | 1.86 | 1.83 | 1.89 | 1.84 |
| w/o First Stage | 2.42 | 2.52 | 2.49 | 2.52 |
| w/o Slow-Fast | 3.51 | 3.55 | 3.48 | 3.55 |
| **Ours** | **4.69** | **4.61** | **4.66** | **4.53** |

Table 6: Kruskal-Wallis Test results on the user study of comparisons with previous works.

| | Preservation | Quality | Consistency | Synchronization |
|---|-------------|---------|-------------|-----------------|
| P-Value | $9.37 \times 10^{-256}$ | $1.73 \times 10^{-193}$ | $1.43 \times 10^{-233}$ | $6.65 \times 10^{-235}$ |

## A.3 FUTURE WORK

Although our method demonstrates strong performance in co-speech gesture video generation, there is significant potential for further improvement. Below, we outline several key areas for future exploration.

**Lip Synchronization.** In talking-head video generation, effective lip synchronization is achieved through either training on large-scale datasets using diffusion models (Tian et al., 2024; Xu et al., 2024a;b) or incorporating specialized lip synchronization modules (Ye et al., 2024; Peng et al., 2024). However, our method does not currently address this aspect, and our dataset is insufficient

Table 7: Kruskal-Wallis Test results on the user study of ablation analysis.

|  | Preservation | Quality | Consistency | Synchronization |
|---|---|---|---|---|
| P-Value | $7.37 \times 10^{-310}$ | $2.33 \times 10^{-297}$ | $5.37 \times 10^{-296}$ | $1.12 \times 10^{-275}$ |

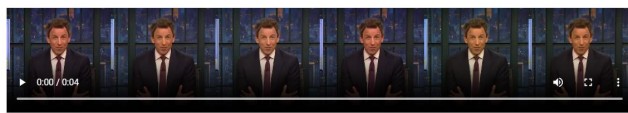

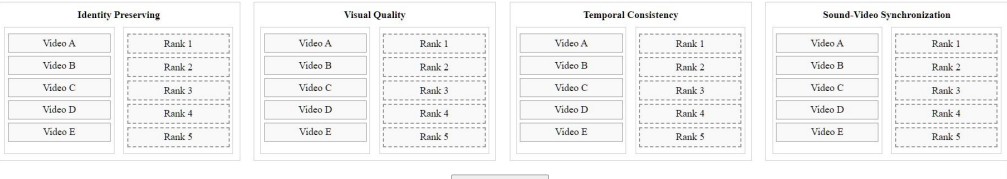

Figure 7: The interface allows users to drag the video ID to the corresponding rank ID, with the option to double-click to cancel the selection. Final rankings are displayed in the result box after clicking "Save Results." We also design an interactive window where incomplete tasks are highlighted in red, enabling users to identify and address any unranked videos easily.

for such training due to its limited size. Improving lip synchronization is a primary focus for future research.

**Larger Dataset.** Although we introduce a new large-scale dataset, it only includes four identities and 33 hours of video. A more comprehensive benchmark is needed for this task. A key question is determining the dataset size required for each identity to accurately generate high-quality videos that replicate individual gesture styles.

**Advanced Attention Mechanism.** Our current approach uses a basic cross-attention mechanism to connect the audio-to-motion and video diffusion branches. Future work could explore more advanced attention mechanisms to better capture and represent expression and pose, thereby enhancing the motion information for the video diffusion process.

**Imbalanced Identities.** Despite containing only four identities, our dataset suffers from an imbalance in data distribution across subjects. This issue requires deeper analysis and effective solutions to ensure a balanced representation of model training.

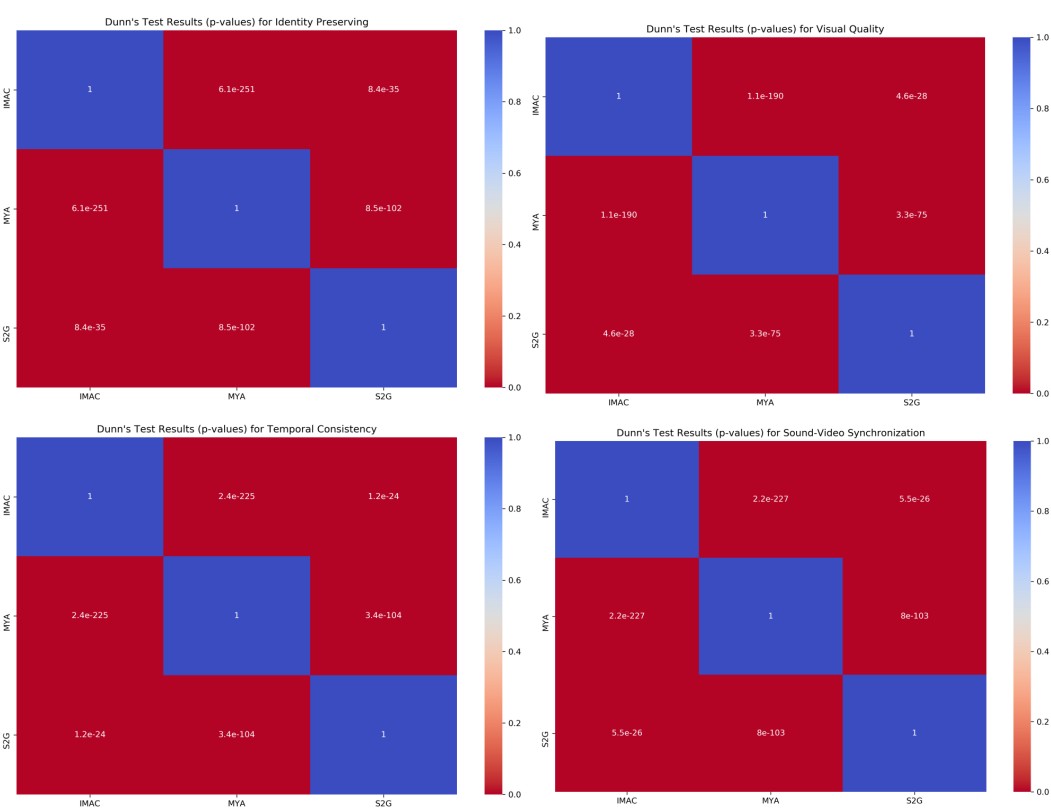

Figure 8: Heat map of pairwise comparisons using Dunn's test for the user study comparing with previous works. Warmer colors (closer to red) indicate greater statistical significance in the differences between models, while cooler colors (closer to blue) denote lower statistical significance. This visualization facilitates the quick identification of the most and least distinct model pairs.

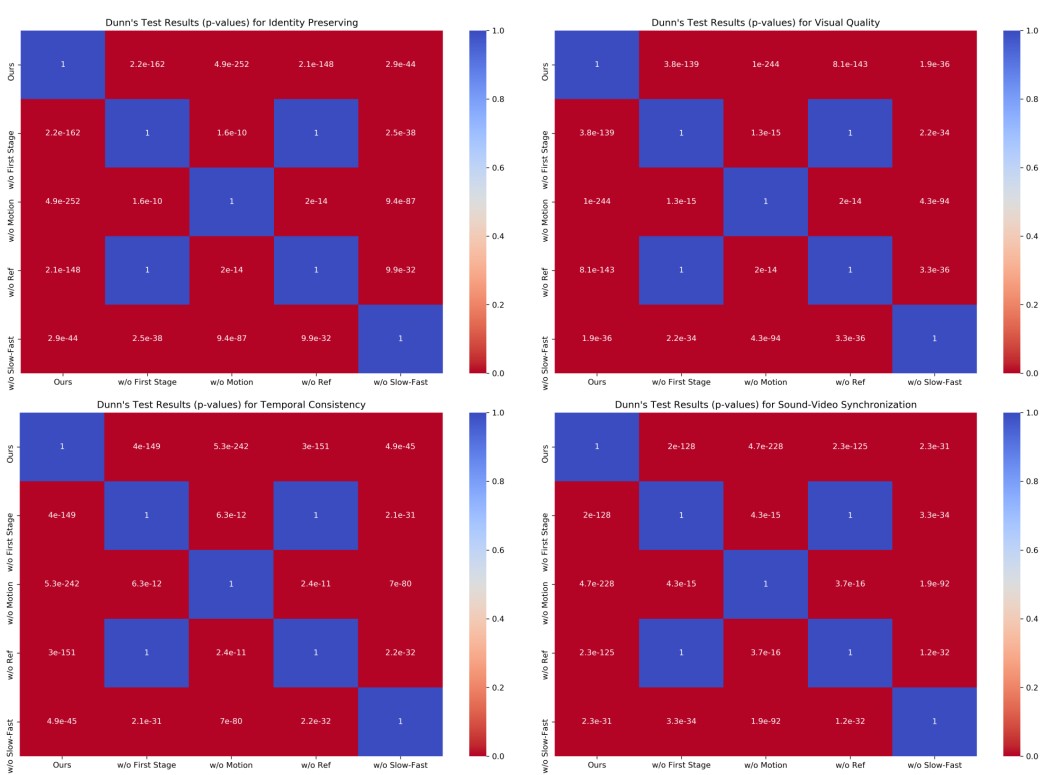

Figure 9: Heat map of pairwise comparisons using Dunn's test for the user study in the ablation study. Warmer colors (closer to red) indicate greater statistical significance in differences between models, while cooler colors (closer to blue) represent lower statistical significance. This visualization allows for quick identification of the most and least distinct model pairs.

