# OpenReview forum: "IMAC: Implicit Motion-Audio Coupling for Co-Speech Gesture Video Generation"
_ICLR.cc/2025/Conference — ICLR 2025 Conference Withdrawn Submission_

### Official Review · Reviewer_ZGTT · 2024-10-15

**Soundness:** 2
**Presentation:** 3
**Contribution:** 2
**Rating:** 3
**Confidence:** 4

**Summary:**

The paper presents an approach for co-speech gesture video generation through the Implicit Motion-Audio Coupling (IMAC) method. The primary contributions of the paper include:

Two-Branch Framework: The IMAC method introduces a two-branch architecture that effectively couples motion parameters with audio inputs, allowing for the generation of gestures given speech and reference image.

Two-Stage Slow-Fast Training Strategy: To address the limitations of existing video diffusion models, which often struggle with capturing long gestures due to memory constraints, the authors propose a two-stage slow-fast training strategy. This approach enables the model to learn from a larger number of frames, enhancing its ability to generate meaningful gestures.

**Strengths:**

1. The image generation qualities are high for hand and shoulder areas.
2. The overall training details of this work looks clear and easy for reimplementation.

**Weaknesses:**

1. This work utilizes Cross-Attention modules and ReferenceNet like structures augmented into the Image Diffusion Model, providing guidance generated from audio and source image. However, completely replying on this strategy does not bring additional novel and inciteful idea within co-speech to gesture generation.It is more likely to bring recent Audio-driven talking head synthesis methods directly to this domain, eg. Hallo, MuseTalk, etc.

2. From the supplementary videos, it is easy to find out that the gesture patterns generated for each video are very limited. There are many repeated gesture patterns. I am concerning whether simply merging the audio features into the Diffusion models will learn the triggers of gestures from the audio.

3. The lip synchronization is a significant problem in the supplementary videos. Even if there is no audio for the speaker but the background audio, the speaker still moves his mouth.

**Questions:**

1. I would like to know how the FGD metrics are conputed for evaluation. Given that there are many repeated gesture patterns but seems not reflected by the evaluation metrics. Specifically, how many frames are used to train the VAE? Some previous works in Gesture generation domain uses 34 frames to train the VAE for evaluation, do the authors follow this pipeline?

2. During the Slow-Fast training, it seems like adding the context window can significantly improve the diversity. I think the limited context window potentially is the main reason for repeated gesture patterns so that the model only learns the beat alignment but ingore the semantics from audios. Will increasing the context window to a longer range further improve this and reduce the repeated motion patterns? (Though adding additional experiments is not a fair thing in rebuttal period, I hope the authors might have some existing experiments to demonstrate there could be a trend, this could be an interesting result to present)

3. During the inference stage, the author use a sliding window for generating long sequences. When moving the the next sliding window and use the last frame generated to serve as the starting frames, will this lead to discontinus problem or unatural transitions of gesture motions?

4. What is the training and inference time for this model compared with the existing methods in details?

---

### Official Review · Reviewer_b4XL · 2024-10-30

**Soundness:** 3
**Presentation:** 2
**Contribution:** 2
**Rating:** 5
**Confidence:** 4

**Summary:**

The paper aims to generate speech-synchronized video from audio input. The authors hypothesize that using explicit visual control signals could degrade video generation quality, so they adopt implicit motion and face representation as control signals to guide the video generation process. However, this hypothesis is not validated through ablation studies. The paper also proposes a two-stage framework, where motion is generated first, followed by a joint training of the motion generation and video generation models. A two-stage slow-fast training strategy is employed to enhance efficiency, and a dataset specifically designed for co-speech gesture video generation is collected.

**Strengths:**

1. The paper is readable, with a well-structured format and clear motivation.

2. In the introduction, the authors primarily aim to address issues such as inconsistent backgrounds, blurry hands, and distorted hands. Based on the comparison videos in the supplementary materials, the method in this paper indeed generates clearer hand movements.

3. The authors present an interesting scientific hypothesis regarding whether explicit visual control might degrade video generation quality. I would like to see further validation of this hypothesis.

**Weaknesses:**

1. Whether explicit visual control truly degrades video diffusion quality requires further investigation and validation. The authors' ablation study does not directly compare this aspect.

2. The pipeline of generating motion and facial expressions based on audio and then using them as control signals for video generation is already widely used.

3.  Why is a new dataset necessary, and how does this dataset differ from previous ones?

4. The experiments are insufficient, lacking comparative results on public datasets, and the number of other methods included in the comparison is too limited.

**Questions:**

1. Considering how to inject control signals is a valuable research question. Why do the authors believe that injecting explicit visual signals would lead to image quality degradation? What are the pros and cons of different control signals, such as pose images, TPS, and 3D meshes?

2. In the two-stage inference process, where a long sequence of motions is first generated and then used as control signals to generate short video clips, is there any temporal discontinuity between video clips? If so, how was this issue addressed?

3. How about the generation efficiency?

---

### Official Review · Reviewer_v71i · 2024-11-02

**Soundness:** 3
**Presentation:** 2
**Contribution:** 2
**Rating:** 5
**Confidence:** 4

**Summary:**

The paper presents a method for generating co-speech gesture videos from input audio and reference images. To achieve this, the authors propose a two-branch training framework and a two-stage training strategy. Of the two branches in their framework, one consists of a VAE-style video diffusion network that can be conditioned with encoded representations of the reference images as well as motion parameters learned from the other branch. This other branch takes in the input audio data and translates those motion parameters through the use of attention mechanisms. In the two-stage training strategy, the first stage trains the audio-to-motion-parameter branch using long sequences of 80 frames, and the second stage additionally trains the video diffusion branch using short sequences of 16 frames sampled randomly from the corresponding long sequences. This particular strategy is aimed to strike a balance between training convergence time and memory load. The authors propose a large-scale video dataset to train and test methods for co-speech gesture video generation, consisting of talk show videos hosted by four individuals. Through quantitative and qualitative comparisons, ablation experiments, and a user study, the authors demonstrate the benefits of their proposed approach.

**Strengths:**

1. The two-branch approach of learning motion parameters from audio and using those to guide video generation is sound and a practical approach to make the video generation problem solvable for various people with their individual gesticulation patterns.

2. The presented experiments and results extensively highlight the benefits of the proposed approach from different perspectives.

**Weaknesses:**

1. While the benefits of the proposed approach are empirically clear, the benefits of the proposed dataset are less clear. Particularly, there are existing datasets focusing on individualized co-speech gesture videos, such as the Berkeley Speech-Gesture Dataset [A], which later had 3D annotations added [B], providing videos of the same talk show hosts among other individuals. What gaps in the usability of existing datasets does the proposed dataset address?

[A] Ginosar, Shiry, Amir Bar, Gefen Kohavi, Caroline Chan, Andrew Owens, and Jitendra Malik. "Learning individual styles of conversational gesture." In Proceedings of the IEEE/CVF Conference on Computer Vision and Pattern Recognition, pp. 3497-3506. 2019.

[B] Habibie, Ikhsanul, Weipeng Xu, Dushyant Mehta, Lingjie Liu, Hans-Peter Seidel, Gerard Pons-Moll, Mohamed Elgharib, and Christian Theobalt. "Learning speech-driven 3d conversational gestures from video." In Proceedings of the 21st ACM International Conference on Intelligent Virtual Agents, pp. 101-108. 2021.

2. Moreover, since the existing dataset [A] is already benchmarked, the lack of experiments on this benchmark makes it harder to get a comprehensive idea of the empirical performance of the proposed approach w.r.t. all existing approaches baselined on this benchmark.

3. The technical component of the paper largely consists of verbal claims and descriptions of the approach, but no rigorous explanations, making the technical discussion particularly hard to follow. For example:

    (a) How does the reference image encoder ensure that identity is preserved for each individual throughout the training process? Is the network trained separately for each individual? If so, does it need to retrained for every new individual?

    (b) What do the high- and low-level features of the reference image look like for the proposed approach (Line 227)? Can the authors give some mathematical and/or visual representations?

    (c) Why and how does explicit visual control lead to suboptimal visual quality (Line 236)? The authors also mention system complexity as an additional drawback of existing approaches. How does their system complexity compare to existing methods?

    (d) How does relying on cross-attention diminish the strength of the audio information in the video diffusion network (Line 252)? Is this provable/demonstrable?

    (e) If (d) is true, why are the authors still using cross-attention to pass the combined audio and motion features into the video diffusion network (Line 258)?

4. Any rigorous analysis of the ablation experiments is missing, making it hard to follow the nature of the empirical benefits of the proposed approach. In Table 3,

    (a) Why are the BAS values of the ablated models so close to the proposed model, while the FGD and FVD values are significantly (almost an order-of-magnitude) worse?

    (b) Why does ablating the first stage training lead to much worse FGD than when ablating the motion parameter branch? If the network is trained with motion parameters, does that not provide better conditioning to the video diffusion branch? How is that negated by the training strategy?

5. The qualitative performance is implausible in some examples --- there are significant body and hand gestures even when there is no audio, causing a visual dissonance. Can the authors explain these implausibilities? Some examples I found (not an exhaustive list): raw+noah_cropped+cmu0000040960_0000907-0001165_final.mp4, raw+noah_cropped+cmu0000041241_0000401-0000561_final.mp4,  videos 1 and 4 of John Oliver in comparisons.html.

6. While not concerns yet, there are multiple ethical components to their work that I recommend the authors address and clarify:

    (a) Have the authors looked at any legal implications of using videos of talk show hosts for training purposes in the present date?

    (b) The authors should carefully delineate the potential harms and misuses of a generative model that synthesizes the likeness of any individual for novel audio, especially given the individuals considered in the dataset are relatively well-known public figures running fact-based shows.

    (c) Since all individuals in the training dataset have the same profession and are all males, the authors should also delineate any additional sources of bias in their model's generative performance.

**Questions:**

1. Line 340-341: Since the individual may appear to have a different size within the selected bounding box across different frames of the same video, do the authors take any steps to normalize the appearance?

2. In their experiments, the authors have defined a long sequence as 80 frames and a short sequence as 16 frames. What are the specific factors (e.g., video frame rate, trade-off between length of the the generated video and its quality) that led to these numbers? How can one modify these numbers for other datasets?

3. A known bottleneck of diffusion-based approaches is the long inference time owing to the use of multiple diffusion time steps. To get a more complete idea of the performance of the proposed approach, it would be helpful to know its inference time compared to other baselines.

4. Apart from their proposed future research directions, there are multiple limitations much closer to the proposed approach that the authors should consider mentioning. For example:

    (a) being limited to single-person videos and 4 individuals,

    (b) all videos having a with a fixed background,

    (c) only generating front-view videos.

---

### Official Review · Reviewer_mNiD · 2024-11-03

**Soundness:** 2
**Presentation:** 3
**Contribution:** 2
**Rating:** 3
**Confidence:** 4

**Summary:**

This paper presents a new framework for co-speech gesture generation that integrates an audio-to-motion branch with a motion-to-video branch. To enhance training efficiency, a two-stage slow-fast training strategy is introduced. Experiments conducted on a large-scale dataset demonstrate the superiority of this method.

**Strengths:**

1. The authors propose to combine the audio-to-motion and motion-to-video branches into a single model.

2. Compared to previous methods, IMAC demonstrates improved background consistency.

**Weaknesses:**

1. The MYA framework already employs SMPL-X parameters (referred to as implicit motion parameters in this paper) as motion conditions. This should not be considered a contribution of this work, as the authors appear to overstate its significance.

2. The dataset utilized in the experiments largely replicates that from "Learning Individual Styles of Gestures," utilizing the same anchors from the same television shows. While the original dataset comprises 144 hours of video, the dataset in this paper is limited to only 33 hours. Therefore, this dataset should not be regarded as either novel or large-scale, and thus does not constitute a contribution to the paper.

3. Upon reviewing the supplementary videos, I observed that the proposed method tends to generate repetitive rhythmic motions, which is undesirable and may indicate overfitting.

4. I also noted that MYA produces consistent backgrounds when the input motion is within a normal range, as demonstrated in the first video of "More Videos for Comparisons." This raises concerns about the quality of the generated motion conditions provided to MYA.

5. The authors should separately compare the audio-to-motion and motion-to-video branches against baseline methods. Specifically, they should input the same data into MYA as used in the proposed method and evaluate the outputs accordingly.

**Questions:**

1. Could the authors present additional results seperately comparing the audio-to-motion and motion-to-video branches against baseline methods?
2. Could the authors provide more videos generated by the proposed method that extend beyond repetitive rhythmic motions?

---

### Note · Authors · 2024-11-12

I have read and agree with the venue's withdrawal policy on behalf of myself and my co-authors.